# Co-Occurrence of *Staphylococcus aureus* and Ochratoxin A in Pasteurized Milk

**DOI:** 10.3390/toxins14100718

**Published:** 2022-10-21

**Authors:** Zhenzhen Zhang, Yanmin Song, Liyan Ma, Kunlun Huang, Zhihong Liang

**Affiliations:** 1College of Food Science and Nutritional Engineering, China Agricultural University, Beijing 100083, China; 2Beijing JTM International Food Co., Ltd., Beijing 101400, China; 3The Supervision, Inspection and Testing Center of Genetically Modified Organisms, Ministry of Agriculture, Beijing 100083, China; 4Beijing Laboratory for Food Quality and Safety, College of Food Science and Nutritional Engineering, China Agricultural University, Beijing 100083, China

**Keywords:** Ochratoxin A, *Staphylococcus aureus*, pasteurized milk, Beijing, AFM_1_

## Abstract

Pathogens and mycotoxins are serious public health risks for humans and food safety in milk. This study concentrated on detecting *Staphylococcus aureus* and Ochratoxin A (OTA) in 210 pasteurized milk from ten urban Beijing districts to suggest the co-occurrence of *S. aureus* with toxin-producing genes and OTA in milk and the possible risk. *S. aureus* was identified by physiological and biochemical experiments and molecular biology experiments, and enterotoxin genes were identified by PCR. OTA was detected by LC-MS/MS. The study found 29 isolates of *S. aureus*, of which 17.24% had the sea gene encoding enterotoxin A. OTA was detected in 31 out of 120 samples and the maximum amount of detection was 18.8 μg/kg. The results of this study indicate that when failing to guarantee the cold chain, the presence of *S. aureus* with enterotoxin genes in milk will present a risk to food safety. Furthermore, the high detection rates and levels of OTA in milk suggest that OTA is a hidden risk. The co-occurrence of *S. aureus* and OTA in milk is a food safety concern and there is a need to control the occurrence of these two biohazards in milk.

## 1. Introduction

Milk is a highly nutritious food containing many macronutrients and micronutrients including proteins, different types of fatty acids and lactose, minerals, antioxidants, and vitamins, that are essential for the growth and maintenance of human health, especially for infants, children, and older adults [1,2]. Therefore, increasing consumption of milk has been observed owing to its high nutritional role in human health throughout the world [3,4]. According to the 2021 China Dairy Quality Report, in 2020, China produced 27.804 million tons of dairy products, and low-temperature milk including pasteurized milk and low-temperature yogurt reached 2.308 million tons, accounting for 8.3%. However, the nutritional richness of milk also makes it susceptible to contamination by microorganisms and toxins [5]. *Staphylococcus aureus* and Ochratoxin A (OTA) are biohazards, which commonly occur in milk and milk products [6]. Milk is an important food source of foodborne illness due to contamination with *S. aureus* [7]. Furthermore, mycotoxin contamination in milk is an emerging concern around the globe [8]. Therefore, it makes sense to test for microorganisms and biotoxins in milk.

Staphylococcal food poisoning (SFP) is one of the most common foodborne diseases worldwide. It is mainly caused by staphylococcal enterotoxins (SEs) secreted by *S. aureus* [9]. SEs that have been found include SEA, SEB, SEC, SED, SEE, etc., which are the main cause of SFP, accounting for more than 90% of global SFP cases. SEA is the most common cause of food poisoning in the United States, accounting for 77.8% of all SFP cases [10]. In China, microbial food poisoning accounted for 53.70% of food poisoning emergencies in 2015. Furthermore, *S. aureus* was an important pathogenic factor in these cases [11]. According to the outbreak reports from 15 European countries, milk and dairy products represented 1–9% of all the incriminated foods in staphylococcal outbreaks [12]. Milk is an important source of SFP. There are several foodborne outbreaks of *S. aureus* intoxications associated with the consumption of contaminated milk [7]. In 1985, there was an outbreak of food poisoning caused by enterotoxin-contaminated milk in a school in Kentucky, and more than 1000 children were affected [13]. In 2000, 13,420 people suffered from food poisoning due to drinking low-fat milk in Osaka, Japan. Eventually, enterotoxin was detected in the milk [14]. In 2007, 166 people were exposed to food poisoning from milk, cacao milk, and vanilla milk, contaminated with staphylococcal enterotoxin in Elementary school, in Austria [15].

SEs are resistant to many environmental conditions, such as high temperatures, low pH [9], freezing, and drying. For instance, crude enterotoxin A remains active at 100 °C for 2 h in broth and at 121 °C for 28 min in mushrooms. SEs are not completely destroyed during pasteurization (15 s at 72 °C) and are considered to be a potential biological hazard. They are also resistant to human proteolytic enzymes and retain their activity in the digestive tract after ingestion [9]. Children will suffer SFP by ingesting as little as 100 ng of SEs, and vulnerable populations may develop staphylococcal food poisoning with a few micrograms of toxin [16]. Therefore, differentiation between virulent and non-virulent strains is significant for evaluating the potential implications of the presence of this microorganism for food safety and public health [17]. The detection of enterotoxin genes has been used to assess the risk of milk and other foods [12]. 

Ochratoxin A (OTA) is one of the most important and deleterious mycotoxins [18], which is a secondary metabolite produced by various *Aspergillus* and *Penicillium* species [19]. A great deal of animal or cell experiments have reported that exposure to OTA can result in various toxicological effects, including teratogenicity, carcinogenicity [20], mutagenicity, hepatotoxicity [21], and especially nephrotoxicity [22]. Different species have different LD_50_. The tests have shown that dogs and pigs are very susceptible, with oral LD_50_ of 0.2 and 1 mg/kg b.w. [23]. Apart from the toxicity of OTA, it is not easy to remove, and can only be destroyed when heated above 250 °C for several minutes [24]. OTA has been reported to extensively occur in feed and food, such as beans, coffee beans, cereals, milk, meat, etc. [5]. Despite efforts to control fungal contamination, extensive mycotoxin contamination has been reported in both developing and developed countries in animal feed [25]. When the animals consume contaminated feeds, one part of the toxin is degraded by bovine rumen microorganisms and the other part remains in the body, resulting in contaminated animal products like egg, milk, liver products etc. [6,26]. Furthermore when the concentration of OTA in the feed is high, there is a high risk of residual OTA in the milk [27]. Few studies have been carried out for monitoring mycotoxins other than AFM1 in milk [28]. OTA has been previously reported in milk and its products [18]. Additionally, a sample contaminated at 2.730 μg/L has been found in Sudan, which indicates public health hazards [29]. For several reasons, it is critical to detect OTA in milk. The first is that OTA is extremely toxic and difficult to remove, the second is that OTA can access cow’s milk through feed, etc. and pose a hidden risk, and the third is that studies to detect OTA in milk are rare and have not received more attention, despite reports of high levels of OTA contamination in milk. Thus, OTA in milk could present an hidden risk. Therefore, testing for OTA in milk is necessary.

Another prominent toxin after aflatoxin is OTA, and *S. aureus* is a common pathogenic bacterium in milk. Pasteurized milk from ten urban districts in Beijing was sampled over the span of a year to evaluate the enterotoxin genes and the fungal toxin OTA as well as the relationship between enterotoxin genes and enterotoxin. As a result, this study focuses on biological risk factors in milk, including *S. aureus* and OTA, with a focus on toxins, to suggest the co-occurrence of *S. aureus* with enterotoxin genes and OTA in milk and the potential risk.

## 2. Results

### 2.1. Identification of S. aureus and Detection of Enterotoxin Genes

Forty-seven isolates of presumed *S. aureus* were isolated in 210 pasteurized milk. Twenty-nine out of 47 isolates were confirmed as *S. aureus* after coagulase, thermonuclear, biochemical tests and Polymerase Chain Reaction technology. As shown in Figure 1, in lanes 1–5, there is a bright band at 592 bp, which is the *nuc* gene. *S. aureus* isolates were further analyzed by PCR for the presence of the *sea*, *seb*, *sec*, *sed*, and *see* genes. The most frequently detected gene was *seb* (7; 24.14%) followed by *sec* (6; 20.69%), *sea* (5; 17.24%), *sed* (4; 13.79%), *see* (3; 10.34%) (Table 1).

### 2.2. Occurrence of OTA in Pasteurized Milk 

Figure 2 and Figure 3 respectively show the LC-MS/MRM chromatograms for the standard OTA and milk samples. In both figures, there is a peak that matches retention times of 3. 667 and 3.749 min, respectively.

The contamination levels of OTA in 120 pasteurized milk were evaluated in this work. The limit of detection (LOD) is 0.015 μg/L and the limit of quantification (LOQ) is 0.049 μg/L. OTA was found in 31 pasteurized milk samples (range 0.11–18.8 μg/L). 25.83% (31/120) of pasteurized milk was contaminated with OTA. 16.13% (5/31) of the samples had a contamination level of more than 10 μg/L (Table 2). Appendix A has provided detailed detect results for the entire year.

In this study, OTA was monitored throughout the year. The results show that the content of OTA detected in winter (October and December) was higher than that in summer (July–September). OTA was not detected from March to May. The content of OTA detected in December reached 18.80 μg/L (Figure 4). 

## 3. Discussion

Our results show the highest detection rate of *seb*. Some studies have also examined the classical enterotoxin genes in milk, with the highest detection rate of *sed* in Bianchi’s study at 25% (120/481) [12] and the highest detection rate of *see* in Grispoldi’s study at 47.06% (8/17) [17]. The most common enterotoxins produced by *S. aureus* isolated from dairy products of bovine or sheep origin were found in the literature to be SEC and SED [30]; the most common enterotoxin produced by *S. aureus* involved in food poisoning outbreaks was SEA [31]. This is probably related to the differences in the ecological reservoir of *S. aureus* in different countries and regions of the world [17]. The presence of enterotoxin-producing isolates of *S. aureus* in pasteurized milk means that failing to guarantee the cold chain could present a food safety risk, particularly if all enterotoxigenic isolates could potentially produce SEA in milk [17]. Research shows that more than half of *S. aureus* isolates contain at least one gene coding enterotoxin, indicating that milk contaminated with *S. aureus* is likely to cause food poisoning [32]. Our study results show that 17.24% of *S. aureus* in pasteurized milk contained *sea* gene. Therefore, pasteurized milk in these ten urban districts of Beijing may have potential food safety risks.

It is generally accepted that SE production constitutes a risk when *S. aureus* bacteria exceed a threshold of 10^5^
*S. aureus* CFU/mL of milk during manufacture [33,34]. For example, the production of enterotoxins SEA and SEB are detected in milk when the count of *S. aureus* exceeds the critical level of 10^5^ CFU/mL [35]. Many studies indicate that temperature and pH might also influence the expression of genes that code for the production of enterotoxins [36,37]. A study shows that the production of SEA can usually be detected at 10–45 °C and the yield of SEA increases with the increasing temperature [33,38]. A study indicated that the conditions for SEA production were pH of > 5.0 and aw of > 0.86 at temperatures of > 15 °C [36]. Undissociated lactic acid (1.6 mM compared to 0.2 mM) was reported to be able to increase SEA production of strain cocktails grown in BHI broth [39]. Another study observed sorbic acid stress (0.15%, pH 5) reduced SEA levels using *S. aureus* Sa17 [40]. In addition, different growth substrates lead to different growth behaviors of bacteria, which can also affect the production of enterotoxins [41]. Therefore, there are a variety of factors that influence the production of toxins.

There are few studies on the detection of OTA in milk because of dietary changes (high concentrate ratios and high feeding levels) that reduce protozoa’s capacity for OTA degradation, rumen microbial communities shift, increasing the likelihood that OTA may contaminate milk despite the fact that the rumen microbiota of cattle can degrade OTA [5]. Table 3 shows the detection of OTA in milk in China and abroad. Compared with our results, all are below our maximum detection. Although there are no regulations in other countries of the European Union for OTA in milk, Slovakia sets a limit of 5 µg/kg for milk [42]. Thus, 25.81% of the samples exceeded the Slovak limit for OTA in milk. Mycotoxin-producing strains in feed can multiply and produce toxins during the growth, harvest, and storage of crops. When cows consume contaminated feed, OTA is left in the milk through metabolism [43]. In addition, differences in climate and animal farming systems in different geographical regions may also lead to differences in OTA levels in milk [2]. In addition, different types of milk, such as organic and conventional milk, may also lead to differences in OTA levels due to differences in processing methods and the nutrients contained [44,45]. Some studies have also shown that pasteurized milk is more contaminated in the cold season than in the warm season. A study conducted by Ansari et al. in 2019 on pasteurized cow milk showed that during the cold seasons of the year compared to the warm seasons pasteurized milk samples were more contaminated [46]. Similar results were found in the study of Mokhtari [45]. In the cold season, due to the high humidity in the forage storage area, the possibility of growth of various fungi, including *A. flavus* and *A. ochraceus* in the forage and forage, will increase, so the contamination rate of OTA will increase. However, in the warmer season, starting around March, dairy farms have access to fresh feed, reducing the OTA content in milk [2]. At the same time, we noticed that the results for November, January, and February were anomalous compared to the results for October and December. The most important reason for this is that the samples from these three months were produced by cows late to feed of better quality. 

Although aflatoxins, especially AFM1 are most commonly found in milk and dairy products in many other countries [1], our study found that AFM1 in milk from the Beijing area was well-controlled in the sample. AFM1 was not detected in 120 pasteurized milk. AFM1 was detected in two samples of 360 UHT milk, and the detected amounts were 0.27 μg/kg and 0.10 μg/kg, respectively. In contrast, OTA was detected in 22.22% (80/360) of UHT milk (Appendix A.). In 2019, it was reported that the mean value of AFM1 and OTA in pasteurized milk was 0.01286 μg/kg and 0.135 μg/kg, respectively [44], which is consistent with our results. OTA is classified as a Group IIB carcinogen to humans by the International Agency for Research on Cancer [52]. A review of studies on OTA over the past 50 years suggests that the carcinogenicity of OTA may also occur in humans [18]. Although no direct evidence of carcinogenicity to the human body has been found at present, the contamination range of OTA is very wide, the contamination level is very high, and its harm is very great. Therefore, research on the real toxicity of OTA should be paid more attention to by more scholars.

Our research results show that OTA is a hidden risk in milk and it serves as a warning and calls attention to the detection of OTA in milk. Several studies in the last two years have examined the prevalence of OTA in different types of milk. A study conducted in 2016 reported OTA levels ranging from 0.34 to 13 μg/L. The detection rate was 80% (32/40) [5]. The detection levels of OTA in this reference were similar to our results. However, the detection rate was even higher than ours. The OTA detection values of several other papers were relatively small. As a result, OTA in milk is not now garnering more attention than before, and high levels of OTA are still detected in milk samples. Consequently, our work still has warning implications. 

## 4. Conclusions

This study monitored *S. aureus* and OTA in pasteurized milk samples in Beijing throughout the year for the first time. The results of this study indicate that when failing to guarantee the cold chain, the presence of *S. aureus* with enterotoxin genes in milk will present a risk to food safety. Furthermore, the high detection rates and levels of OTA in milk suggest that OTA is a hidden risk. As a result, the findings of this study have some bearing and can be used as a reference point for biological risk factors in milk.

## 5. Materials and Methods

### 5.1. Sampling

From October 2014 to September 2015, a total of 210 pasteurized milk (including 90 copies of Brand A and 120 copies of Brand B) were bought from supermarkets located in 10 urban districts in Beijing, China. (Figure 5). All samples were delivered at 4 °C and analyzed within 24–48 h. In total, 210 samples of pasteurized milk were detected for *S. aureus*, and only 120 samples of Brand B were detected for OTA.

### 5.2. Isolation and Detection of S. aureus

We carried out the culture and identification of *S.*
*aureus* according to the methods described by GB 4789.10-2016. The samples were cultured at 37 °C and 200 rpm for 12 h for 18 h, then crossed on the selective Baird-Parker plate and cultured at 37 °C for 48 h. *S. aureus* colonies on B-P plates were round, 2–3 mm in diameter, gray or black in color, and surrounded by a turbid zone. The suspected colonies were selected for Gram staining and plasma coagulase test. Gram staining microscopic examination showed that *S. aureus* was Gram-positive cocci. The experiment on plasma coagulase is as follows. A single suspicious colony was picked from a Baird-Parker plate, inoculated into 5 mLBHI broth, and incubated at 37 °C for 18 h. In the ultra clean table, 0.5 mL of saline was added to the lyophilized rabbit plasma, shaken to dissolve it, then 0.2–0.3 mL of BHI culture was added, shaken well, placed in 37 °C incubator, and observed every half hour for 6 h. The positive result was determined if the volume of clotting or clotting was greater than half of the original volume. The broth culture of the positive plasma coagulase test was also used as the control. 

Then the suspected colonies to increase the bacteria and extract the genome were picked out. DNA was extracted with the TIANamp Bacteria DNA Kit (OSR-M502, Tiangen, China). The *nuc* gene acts as a marker and also the presence of heat resistant nuclease gene (*nuc*) is strongly associated with the production of enterotoxin and it can be considered an indicator of infection with enterotoxin producer *S. aureus* [14]. Therefore, the *nuc* gene was amplified by PCR to identify the *S. aureus*. PCR was also used to detect the presence of the classic enterotoxin genes *sea*–*see*. The PCR reaction was conducted on a C1000 Toucah Thermal Cycler (Bio-Rad, Johannesburg, South Africa). The cycling conditions were initial denaturation at 94 °C for 5 min; followed by 30 cycles of 94 °C for 30 s, 64 °C for 30 s, 72 °C for 60 s, and a final elongation step of 72 °C for 10 min. The PCR products were stored at 4 °C and later separated by 1% agarose gel electrophoresis. D2000 DNA Marker was used. The PCR primers were designed with NCBI according to the nuclease gene sequence, as shown in Table 4. (GenBank: V01281.1, http://www.ncbi.nlm.nih.gov/nuccore/V01281.1, accessed on 2 March 2015).

### 5.3. Detection of OTA

#### 5.3.1. OTA Extraction

We extracted OTA from samples according to the methods described by the GB 5009.96-2016. Pipette 5 mL of fresh milk into 50 mL centrifuge tube, add 20 mL of acetonitrile (84%), add 2 g of anhydrous magnesium sulfate and 1 g of sodium chloride, vortex 2 min, ultrasonic 20 min, and centrifuge at 1650 g for 5 min, 10 mL of the supernatant was evaporated dry in a rotary evaporator at 60 °C. Dissolve with 1mL of methanol, then add 1mL of water, mix well, and pass through a 0.22 μm filter membrane, to be analyzed. Each sample was analyzed in triplicate. OTA standard was purchased from J&K Scientific Ltd. Beijing, China.

#### 5.3.2. LC-MS/MS Analysis 

The LC-MS/MS analysis was performed as previously reported [47]. The sample extracts were analyzed in an isocratic elution with an Agilent Pro shell 120 EC C_18_ column (50 mm × 2.1 mm, 3.5 μm; Agilent Technologies, Little Fall, DE, USA) by using Agilent 1260 Infinity Quaternary LC system. The mobile phase consisted of (A) 0.1% formic acid aqueous solution and (B) Acetonitrile. The following linear gradient program was used: 20% B in 0–1 min; 20–65% B in 1–5 min; 65–85% B in 5–8 min; 85% B in 8–10 min; 85–20% B in 10–10.1 min; 20% B in 10.1–16 min. The flow rate was 0.3 mL/min while the injection volume was 10 μL.

Following separation, the column effluent was connected to a triple-quadrupole mass spectrometer (Agilent Technologies 6460, CA, U.S.A.) equipped with an ESI source. OTA was detected in positive mode using MRM. Data acquisition and mass spectrometric evaluation were carried out on a Mass Hunter Workstation (Agilent Technologies, Santa Clara, CA, USA). Operating conditions were: Spray Voltage–5500 V, Curtain Gas–30 psi, Temperature–500 °C, Gas 1–30 psi, Gas 2–50 psi, Entrance Potential (EP)–10 V, Collision Energy (CE)–21 eV. De-clustering Potential (DP)–61 V, Exit Potential (CXP)–20 V. The precursor ion was monitored and collision induced dissociation was used to generate product ions. The precursor ion was m/z 404, the product ion was m/z 358 and m/z 238.8, and the ion ratio was 1:1. To evaluate the linearity, five-point calibration curves were constructed to calculate the determination coefficients (R2). The signal-to-noise (S/N) approach was used to estimate the LOD and the LOQ. The gradient OTA standard solution was injected on the liquid chromatograph. The LOD and LOQ were defined based on signal (S)-to-noise (N) ratios of S/N > 3 and S/N > 10, respectively. 

LC-MS/MS analysis was done at the Supervision, Inspection & Testing Center for Agricultural Products Quality.

#### 5.3.3. Statistical Analysis

The data were analyzed by SPSS. The significance level was set at *p* < 0.05, and all experiments were replicated at least three times.

## Figures and Tables

**Figure 1 toxins-14-00718-f001:**
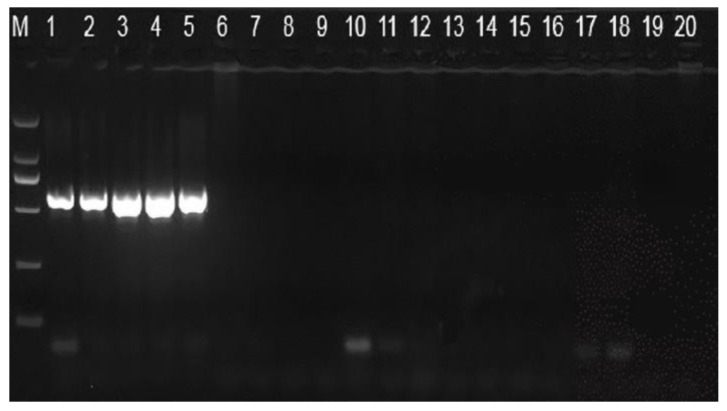
PCR amplification specificity detection of the *nuc* gene. M: D2000 marker; 1: *Staphylococcus aureus* ATCC25923; 2: *S. aureus* ATCC6538; 3: *S. aureus* CGMCC 1.89; 4: *S. aureus* CICC10786; 5: *S. aureus* MW2; 6: *Salmonella*; 7: *Pseudomonas aeruginosa*; 8: *Bacillus cereus*; 9: *B. amyloliquefaciens*; 10: *Lactobacillus rhamnosus*; 11: *Lactobacillus*; 12: *L. Casei*; 13: *S. lentus*; 14: *S. haemolyticus*; 15: *S. Arlette*; 16: *S. epidermidis*; 17: *S. chromogenes*; 18: *S. cohnii*; 19: *S. sciuri*; 20: *S. saprophyticus*.

**Figure 2 toxins-14-00718-f002:**
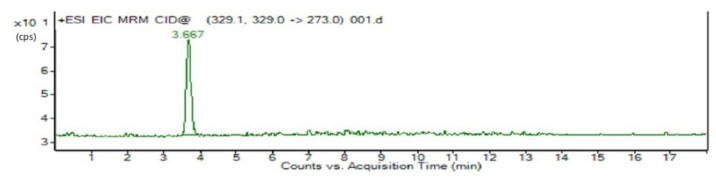
Chromatogram of OTA standard detection.

**Figure 3 toxins-14-00718-f003:**
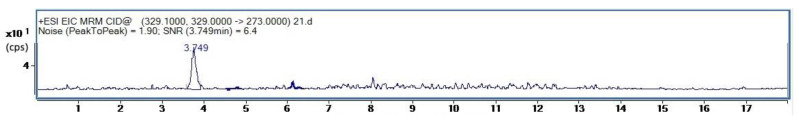
Chromatogram of OTA detection in pasteurized milk.

**Figure 4 toxins-14-00718-f004:**
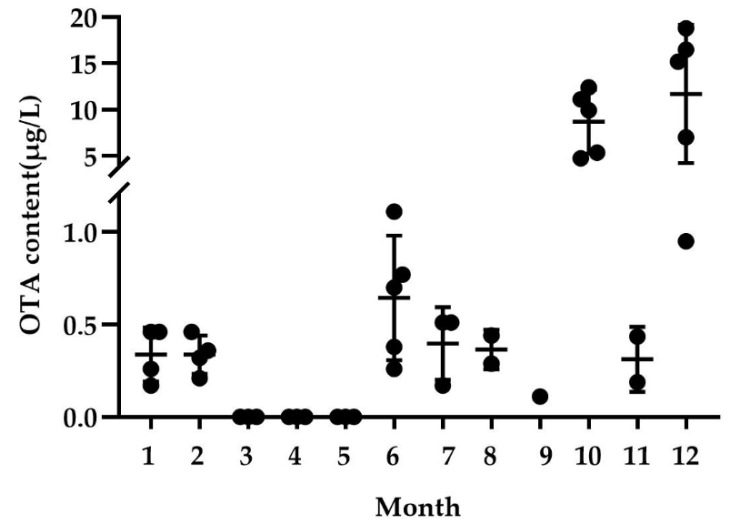
OTA-detection level in each month in pasteurized milk. Data from December 2014 to September 2015.

**Figure 5 toxins-14-00718-f005:**
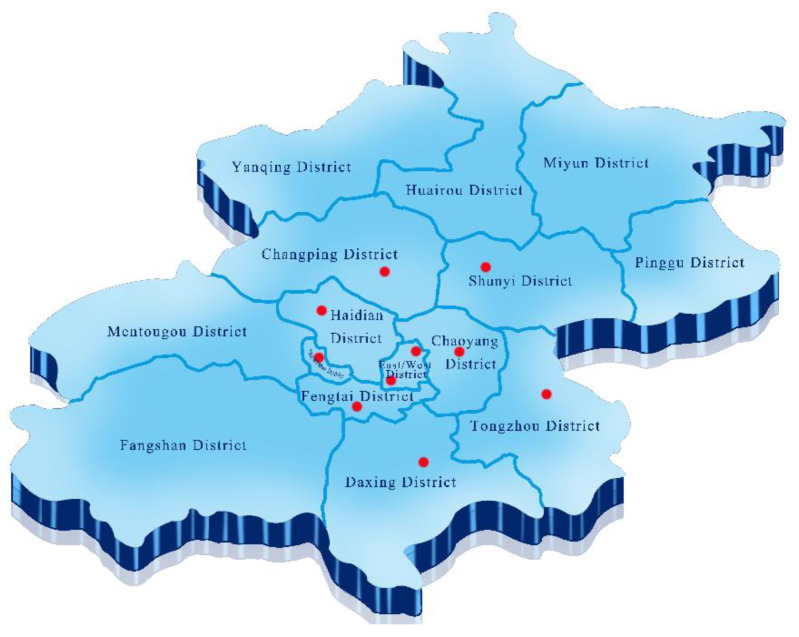
Sampling location map. The red dot represents the actual sampling location.

**Table 1 toxins-14-00718-t001:** Detection of enterotoxin genes of *S. aureus*.

Enterotoxin Gene	*sea*	*seb*	*sec*	*sed*	*see*
Number	5	7	6	4	3
Proportion (%)	17.24	24.14	20.69	13.79	10.34

**Table 2 toxins-14-00718-t002:** Contamination level of OTA in pasteurized milk.

	Contamination Level of OTA (μg/L)				Total
	>0.049 and <1.0	1.0–5.0	5.0–10.0	>10.0	
Number	21	2	3	5	31
Proportion (%)	17.50	1.67	2.5	4.16	25.83

**Table 3 toxins-14-00718-t003:** Occurrence of OTA in milk at home and abroad.

Country	Sample	Method of Analysis	LOD μg/kg	LOQ μg/kg	Prevalence (%)	Range (μg/L or μg/kg	Reference
China	raw cow milk	UHPLC-MS/MS	0.004	0.012	-	0.0567–0.0841	[47]
	liquid cow milk		0.003	0.009		0.0268–0.0579	[47]
Italy	organic	LC-FD	-	0.05	3/63 (4.8%)	0.07–0.11	[48]
Sudan	raw cow milk	HPLC-UV	-	-	1/5 (20%)	0.000–2.730	[29]
France	raw cow milk	LC-FLD	-	-	3/264 (1.1%)	0.005–0.0066	[49]
Sweden	raw cow milk	HPLC-FD	-	-	5/36 (14%)	0.010–0.040	[50]
Norway	organic	LC-FLD	-	-	5/47 (11%)	0.015–0.028	[51]
	conventional				6/40 (15%)	0.011–0.058	[51]

“-”: Not detected.

**Table 4 toxins-14-00718-t004:** Primers used in the detection of *S. aureus* and enterotoxin genes.

Gene	Primer	Sequence (5’-3’)	Size (bp)	Annealing Temperature (°C)	Reference
*nuc*	NUC-F	AGGGCAATACGCAAAGAGGTT	592	62	This work
	NUC-R	TGAATCAGCGTTGTCTTCGC			
*sea*	SEA-F	TTGGAAACGGTTAAAACGAA	120	50	[53]
	SEA-R	GAACCTTCCCATCAAAAACA			
*seb*	SEB-F	TCGCATCAAACTGACAAACG	478	50	[53]
	SEB-R	GCAGGTACTCTATAAGTGCC			
*sec*	SEC-F	GACATAAAAGCTAGGAATTT	257	50	[53]
	SEC-R	AAATCGGATTAACATTATCC			
*sed*	SED-F	CTAGTTTGGTAATATCTCCT	317	50	[53]
	SED-R	TAATGCTATATCTTATAGGG			
*see*	SEE-F	TAGATAAAGTTAAAACAAGC	170	50	[53]
	SEE-R	TAACTTACCGTGGACCCTTC			

“5’” and “3’” stand for the DNA sequence’s 3’ and 5’ ends, respectively.

## Data Availability

Not applicable.

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
