# Peer review of "Co-Occurrence of *Staphylococcus aureus* and Ochratoxin A in Pasteurized Milk"

_toxins, 2022, doi:10.3390/toxins14100718_

Round 1
Reviewer 1 Report
In LC-MS/MS Analysis were not indicate precursor ions, product ions, ion ratio and instrumental conditions as collision energy. No information about calibration curve was present.
How was the LOQ calculated?
Figure 2 showed a different retention time between OTA standard and sample not explained (it depend of the matrix??Why data wasn't show for OTA standard?)
Line 101-106 More explications about the climatic influence are necessary and also appropiate references.
Statistical analysis wasn't properly explained.
Author Response
Manuscript ID: toxins-1916153
The line numbers mentioned below refer to those in the revised manuscript. Point by point responses to editor/reviewers:
Reviewer #1:
We are very grateful to the Reviewer #1 for giving us very professional and specific suggestions to revise our manuscript.
Q1. In LC-MS/MS Analysis were not indicate precursor ions, product ions, ion ratio and instrumental conditions as collision energy. No information about calibration curve was present.
Thanks for the suggestion. These instrumental conditions you mentioned are important. The precursor ion is m/z 404, the product ion is m/z 358, m/z is 238.8, ion ratio is 1:1, and Collision Energy (CE) is 21 eV. In addition, we add two other parameters, the De-clustering Potential (DP) and the Exit Potential (CXP). All parameters mentioned above have also been added in the manuscript. (At line 381-383).
In this experiment, we constructed the calibration curve. The method of constructing calibration curve has been supplemented in the method section (At line 383). The calibration curve was y=1382.9x+52.13 with an R2 of 0.9999, indicating a good linear relationship.
Q2. How was the LOQ calculated?
Thanks for the suggestion. We determined LOD and LOQ by using Agilent 1260 Infinity Quaternary LC system and a triple-quadrupole mass spectrometer (Agilent Technologies 6460, CA, U.S.A.). The signal-to-noise (S/N) approach was used to estimate the LOD and the LOQ. The gradient OTA standard solution was injected on the liquid chromatograph. The LOD and LOQ were defined based on signal (S)-to-noise (N) ratios of S/N > 3 and S/N > 10, respectively.
And the methods for the determination of LOD and LOQ as well as the results have been added to the manuscript (At line 384-387).
Q3. Figure 2 showed a different retention time between OTA standard and sample not explained (it depend of the matrix??Why data wasn't show for OTA standard?)
Thank you for the suggestion. The chromatogram of the OTA standard has been added to the manuscript. (At line 138)
After double check, the retention time for the OTA standard and sample is confirmed as 3.667 and 3.749 minutes, respectively. We modified the retention time of OTA in the samples to 3.749 from 3.818 or 3.835 , which was determined after aligning with the excitation and emission wavelengths of the OTA standard. The inconsistency between the retention time of the sample and the standard may be due to a change in column temperature or the pH of the sample matrix affecting the pH of the mobile phase, thus causing the retention time of the sample to fluctuate (At line 102-107).
Q4. Line 101-106 More explications about the climatic influence are necessary and also appropiate references.
Thanks for the suggestion. Two appropiate references[38][39] are cited in the paper to add the effect of climatic conditions on OTA content in the samples. That is, cold season are more likely to contaminate OTA in samples than spring months when everything is revived, mainly due to the different growth conditions of OTA-producing fungi in feeds fed to cows under different climatic conditions, resulting in differences in OTA content in feeds (At line 214-221).
The index of the newly added references is as follows:
[1] Mohammedi-Ameur, S.; Dahmane, M.; Brera, C.; Kardjadj, M.; Ben-Mahdi, M.H. Occurrence and Seasonal Variation of Aflatoxin M1 in Raw Cow Milk Collected from Different Regions of Algeria. Vet World 2020, 13, 433–439, doi:10.14202/vetworld.2020.433-439.
[2] Mokhtari, S.A.; Nemati, A.; Fazlzadeh, M.; Moradi-Asl, E.; Ardabili, V.T.; Seddigh, A. Aflatoxin M1 in Distributed Milks in Northwestern Iran: Occurrence, Seasonal Variation, and Risk Assessment. Environ Sci Pollut Res 2022, 29, 41429–41438, doi:10.1007/s11356-021-18212-9.
Q5. Statistical analysis wasn't properly explained.
Thanks for the suggestion. The conditions, standard curve, detection line, quantification line and retention time about the LC-MS/MS were not clear and caused confusion , and all of them have been revised. First of all, this study mainly uses the instruments of the testing center, which carries out the data all year round, and the reliability of the data is not in question. In addition, the effect of climate on OTA content has been explained in the manuscript. Other data and results were also checked and confirmed throughout the manuscript.

Reviewer 2 Report
This manuscript aims to determine S. aureus and OTA in pasteurized milk collected in Beijing, China.
This manuscript is well designed and written. Here are minor suggestions. 1. Page 4, line 131 -> revise the sentence; missing subject 2. Page 5, line 191 -> what do you mean "normal saline" 3. Page 5, line 192 -> put the unit for 0.2-0.3 of BHI culture4. Page 6, line 222 -> Although you referred to the previous paper, please add LOD, LOQ, and recovery for OTA determination here.
Author Response
Manuscript ID: toxins-1916153
The line numbers mentioned below refer to those in the revised manuscript. Point by point responses to editor/reviewers:
Reviewer #2:
We are very grateful to the Reviewer #2 for giving us very professional and specific suggestions to revise our manuscript.
Q1. Page 4, line 131 -> revise the sentence; missing subject
Thanks for the suggestion. The subject has been added in the manuscript (At line 180).
Q2. Page 5, line 191 -> what do you mean "normal saline"
Thanks for the suggestion. "Normal saline" is used to indicate saline. We made a mistake at the time, and now the translation has been corrected to “saline” according to the standard in the manuscript (At line 310).
Q3. Page 5, line 192 -> put the unit for 0.2-0.3 of BHI culture
Thanks for the suggestion. This phrase was changed to 0.2-0.3 mL of BHI culture in the manuscript (At line 311).
Q4. Page 6, line 222 -> Although you referred to the previous paper, please add LOD, LOQ, and recovery for OTA determination here.
Thanks for the suggestion. We determined LOD and LOQ by using Agilent 1260 Infinity Quaternary LC system and a triple-quadrupole mass spectrometer (Agilent Technologies 6460, CA, U.S.A.). The signal-to-noise (S/N) approach was used to estimate the LOD and the LOQ. The gradient OTA standard solution was injected on the liquid chromatograph. The LOD and LOQ were defined based on signal (S)-to-noise (N) ratios of S/N > 3 and S/N > 10, respectively. And the methods for the determination of LOD and LOQ as well as the results have been added to the manuscript (At line 384-387).
LC-MS/MS Analysis relied on Supervision, Inspection & Testing Center for Agricultural Products Quality. Relying platforms have been added in the method. the detection of mycotoxin belongs to the testing center's regular items, including animal, plant, food raw materials of mycotoxins (OTA, AFM) capability verification, so the results of the method testing is reliable.
The recovery rate of milk is a routine test in the testing center, so the recovery rate was not tested in this experiment. For example, the Testing Center obtained recoveries using LC-MS as in the range of 97.8–103.6% in 2016 which confirmed its high accuracy [1].
References:
[1] Zhao, Y., Huang, J., Ma, L., Liu, S., & Wang, F. (2017). Aflatoxin B1 and sterigmatocystin survey in beer sold in China. Food Additives & Contaminants: Part B, 10(1), 64-68.

Reviewer 3 Report
Dear Authors,
Please consider the following suggestion, in order to improve the submitted work:
General:
· cited references are not always adequate (e.g. Line 25 and 27) or are missing (e.g. line 29)
· formatting: line 72 & line 95 - please start a sentence with numbers in full, not numeric
· English: please revise (e.g. lines 192-195); “Weigh 5 mL” line 215; Line 131: “Compared with our results, are lower than our maximum detection”; line 162 “OTA in milk is not only a potential source of human 162 intake of OTA,”
Detailed:
line 92 – 120 or 210 samples?
Line 93 – please indicate the precise number
Line 100 – winter does not include January, February and March? Is so, description is not correct
Line 102-106 – this should be part of the discussion section
Line 106 – but in January, producers rely on stored feed, silage; why the reduction in the levels of OTA? In addition a characterization of the temperature and humidity that characterize the sampling regions should be provided;
Line 111 – the European regulation does not “show”; a different reference should be used
Line 116 – please cite the study
Line 120 – “failing to guarantee the cold chain” instead of “temperature abuse of the milk”
Line 130 and 136 – contamination and not pollution
Line 134 - were there any differences detected between regions, production types (e.g. organic vs conventional), brands?
Line 140 – please present the results of the validation process in section “Materials and methods”; such level of variation should not be permitted
Line 141 – please refer that given the clear evidence for the carcinogenic effects of OTA, no TDI exist (the previously established TWI of 120 ng OTA/kg bw (EFSA, 2006), so TDI can only be applied for comparison purposes.
Line 141-146 – please notice that such estimation must be supported by a detailed description in the materials and methods sections, which is absent
Table 3 – please add information on the LOD/LOQ of each cited study
Line 150 – please clarify: the present study determined AFM1?
line 208 – concentration of agarose gel is missing
Line 214 – please indicate if GB 214 5009.96-2016refers to a food safety standard from the country
Line 219 - “μm” instead of “um”
figure 4 – image with low resolution, like the remaining ones; bigger images should be presented
Author Response
Manuscript ID: toxins-1916153
The line numbers mentioned below refer to those in the revised manuscript. Point by point responses to editor/reviewers:
Reviewer #3:
We are very grateful to the Reviewer #3 for giving us very professional and specific suggestions to revise our manuscript.
Q1. cited references are not always adequate (e.g. Line 25 and 27) or are missing (e.g. line 29)
Thanks for the suggestion. Thank you for your suggestion. We have enriched references [2][4]. (At line 26-28).
The missing reference is the 2021 China Dairy Quality Report, which I did not find a downloadable channel, so it is not cited as a reference in the text, but the source of the data is added in the text.(At line 31).
The index of the newly added references is as follows:
- Mohammedi-Ameur, S.; Dahmane, M.; Brera, C.; Kardjadj, M.; Ben-Mahdi, M.H. Occurrence and Seasonal Variation of Aflatoxin M1 in Raw Cow Milk Collected from Different Regions of Algeria. Vet World2020, 13, 433–439, doi:10.14202/vetworld.2020.433-439.
[2] Bahrami, R.; Shahbazi, Y.; Nikousefat, Z. Aflatoxin M1 in Milk and Traditional Dairy Products from West Part of Iran: Occurrence and Seasonal Variation with an Emphasis on Risk Assessment of Human Exposure. Food Control 2016, 62, 250–256, doi:10.1016/j.foodcont.2015.10.039.
Q2. formatting: line 72 & line 95 - please start a sentence with numbers in full, not numeric
Thanks for the suggestion. It has been corrected in the manuscript. And the full manuscript is also checked and revised(At line 120-121 & line 143). Thanks again for your suggestions.
Q3. English: please revise (e.g. lines 192-195); “Weigh 5 mL” line 215; Line 131: “Compared with our results, are lower than our maximum detection”; line 162 “OTA in milk is not only a potential source of human 162 intake of OTA,”
Thanks for the suggestion. They have been corrected in the manuscript. They are each modified to “Pipette 5 mL”(At line 358)、“Compared with our results, all are below our maximum detection. ” (At line 182) and “Our findings suggest that OTA is a hidden risk in milk and is a potential food source for our exposure to OTA” (At line 242).
Q4. line 92 – 120 or 210 samples?
Thanks for the suggestion. In fact, we have completed 210 tests of golden glucose in pasteurized milk and 120 tests of OTA in pasteurized milk. The description has been made clear in the methods section of the manuscript (At line 296-297).
Q5. Line 93 – please indicate the precise number
Thanks for the suggestion. This is indeed 120 samples of pasteurized milk.(At line 142).
Precise numbers have been explained in the methods section. One hundred and twenty samples were used to detect OTA and 210 samples were used to detect S. aureus (At line 296-297).
Q6. Line 100 – winter does not include January, February and March? Is so, description is not correct
Thanks for the suggestion. You are correct in understanding that the cold season does include January and February.The average temperature in Beijing in January and February is a few degrees below zero, and according to its climate characteristics, it is known that January and February are included in the range of the cold season. Therefore, when analyzing the level of OTA, the results of January and February should be analyzed. It is obvious that the detected values in January, February and November differ significantly from those in October and December. The most likely reason is that the OTA levels of the feed is different, resulting in larger differences in the OTA levels in the cow, which affects the OTA levels in the milk. The specific interpretation has been added to the manuscript (At line 221-225).
Q7. Line 102-106 – this should be part of the discussion section
Thank you for your suggestion. This section has been moved to the discussion section, and the content has been added and improved (At line 221-225).
Q8. Line 106 – but in January, producers rely on stored feed, silage; why the reduction in the levels of OTA? In addition a characterization of the temperature and humidity that characterize the sampling regions should be provided;
Thank you for the advice. Thanks for the suggestion. Based on the trend of the manuscript results, it would make more sense that the OTA detection should be higher in January and February than it is now. But, our results reflected the reality of the situation. The most likely reason current result is that the samples collected in January and February were of better quality, so the level of OTA detected was lower. The initial experimental design of this manuscript did not take into account the correlation between feed source and sampling location, which is an area that needs to be strengthened in the future (At line 221-225).
We collected the milk samples directly from the supermarket, so the temperature and humidity of the sampling area were not recorded. If we can trace the temperature and humidity to the sampling area is best. Thank you again!
Q9. Line 111 – the European regulation does not “show”; a different reference should be used
Thanks for the suggestion. We confirmed that the European Commission Regulation does not show the number of colonies required to cause the disease. We have cited the new reference as you suggested[30][31] and confirmed that the new citation contains the quote (At line 164-165).
The index of the newly added references is as follows:
[1]Babić, M.; Pajić, M.; Radinović, M.; Boboš, S.; Bulajić, S.; Nikolić, A.; Velebit, B. Effects of Temperature Abuse on the Growth and Staphylococcal Enterotoxin A Gene ( Sea ) Expression of Staphylococcus Aureus in Milk. Foodborne Pathogens and Disease 2019, 16, 282–289, doi:10.1089/fpd.2018.2544.
[2] Duquenne, M.; Fleurot, I.; Aigle, M.; Darrigo, C.; Borezée-Durant, E.; Derzelle, S.; Bouix, M.; Deperrois-Lafarge, V.; Delacroix-Buchet, A. Tool for Quantification of Staphylococcal Enterotoxin Gene Expression in Cheese. Appl Environ Microbiol 2010, 76, 1367–1374, doi:10.1128/AEM.01736-09.
Q10. Line 116 – please cite the study
Thank you for your suggestion. “The study” is about what is described in the next sentence. I'm sorry that the incoherence between sentences has confused you. We have changed these two sentences into one sentence to make the meaning more concise. (At line 169-170)..
Q11. Line 120 – “failing to guarantee the cold chain” instead of “temperature abuse of the milk”
Thanks for the suggestion. It has been corrected in the manuscript (At line 171-172)..
Q12. Line 130 and 136 – contamination and not pollution
Thanks for the suggestion. It has been corrected in the manuscript (At line 239& line 213). The full text was also checked throughout.
Q13. Line 134 - were there any differences detected between regions, production types (e.g. organic vs conventional), brands?
Thanks for the suggestion. According to the literature, regions, product types and brands can also influence the level of OTA in milk. Therefore, these three factors have been added to the manuscript and references have been cited [38][39].(At line 187-213) In addition to the above factors, variations of OTA levels in milk may be related to different methods of detecting toxins, as well as to differences in feed and feed storage quality, and climatic and seasonal variations . All of the above reasons are explained in the manuscript, and references are cited.
The index of the newly added references is as follows:
[1] Mohammedi-Ameur, S.; Dahmane, M.; Brera, C.; Kardjadj, M.; Ben-Mahdi, M.H. Occurrence and Seasonal Variation of Aflatoxin M1 in Raw Cow Milk Collected from Different Regions of Algeria. Vet World 2020, 13, 433–439, doi:10.14202/vetworld.2020.433-439.
[2] Mokhtari, S.A.; Nemati, A.; Fazlzadeh, M.; Moradi-Asl, E.; Ardabili, V.T.; Seddigh, A. Aflatoxin M1 in Distributed Milks in Northwestern Iran: Occurrence, Seasonal Variation, and Risk Assessment. Environ Sci Pollut Res 2022, 29, 41429–41438, doi:10.1007/s11356-021-18212-9.
Q14. Line 140 – please present the results of the validation process in section “Materials and methods”; such level of variation should not be permitted
Thanks for the suggestion. It is true that the results of the November data are rather abnormal, but we had verifide the instrument and the method by calibration curve, LOD, LOQ and recovery and we can tell: the method and the instrument are normal. The most likely reason for this result is the better quality of the batch sample and the lower OTA content in the feed fed to the cows.(At line 221-225).
Q15.Line 141 – please refer that given the clear evidence for the carcinogenic effects of OTA, no TDI exist (the previously established TWI of 120 ng OTA/kg bw (EFSA, 2006), so TDI can only be applied for comparison purposes.
Thanks for the suggestion. We strongly agree with the professional information you gave us. It is indeed inappropriate to compare the TDI value of OTA with our detection here. Therefore, we decided to remove this section on the risk assessment to avoid misleading readers.
Q16. Line 141-146 – please notice that such estimation must be supported by a detailed description in the materials and methods sections, which is absent
Thanks for the suggestion. We have deleted the content about risk assessment here to avoid misleading readers. Therefore, there is no need to describe the method of risk assessment in the materials and methods sections.
Q17. Table 3 – please add information on the LOD/LOQ of each cited study
Thanks for the suggestion. The LOD and LOQ values for each sample have been added to the table according to the values in the references (At line 226). However, some references do not show LOD or LOQ values. Therefore, we use "-" in the table to indicate.
Q18. Line 150 – please clarify: the present study determined AFM1?
Thanks for the suggestion. The exposure of OTA in pasteurized milk were mainly presented in the manuscript. Moreover, AFM1 in pasteurized milk had been detected in pasteurized milk and UHT milk. Detailed data were placed in an supplementary materials. (At line 230-233)
Q19. line 208 – concentration of agarose gel is missing
Thanks for the suggestion. The concentration of agarose gel has been added in the manuscript. (At line 326)
Q20. Line 214 – please indicate if GB 214 5009.96-2016refers to a food safety standard from the country
Thanks for the suggestion. GB 5009.96-2016 is China's national food safety standard, which specifies the method for the determination of ochratoxin A in food. (At line 357-358). In addition, you mentioned “GB 214 5009.96-2016”, which should be added more line number
Q21. Line 219 - “μm” instead of “um”
Thanks for the suggestion. It has been corrected in the manuscript. (At line 362). The full text was also checked throughout.
Q22. figure 4 – image with low resolution, like the remaining ones; bigger images should be presented
Thanks for the suggestion. We have replaced Figure 4 with a higher resolution image. (At line 299)
Reviewer 4 Report
Journal: Toxins (ISSN 2072-6651)
Manuscript ID:toxins-1916153
Type: Article
Title
Co-occurrence of Staphylococcus aureus and Ochratoxin A in Pasteurized Milk
Suggestions:
Rewrite phrase “Therefore, S. aureus containing enterotoxin genes in pasteurized milk could present a food safety risk and needs to be continuously monitored.”
-Therefore, pasteurized milk could be a risk due to the occurrence of toxigenic strains of S…
-I believe the MS should be proofread.
-Line 46: Correct 100 ºC… 121 ºC…
-Line66: I think authors could explain better the necessity of monitoring OTA.
-The authors could restructure the introduction section, adding paragraphs.
-Avoid references that are 10 years old.
-Line 93 how the authors determine LOQ?
-Table 2: Include information about contamination of each 31 sample.
-Figure 3: the quality of the diagram is questionable. Perhaps a table could be more explanatory.
- Since the data is from December 2014 to September 2015, I think the authors should explain why they believe the results are still relevant to the present time. Do they think the situation has changed after 7 years? Include this information in the discussion section.
-line 102: It is a logical sentence, but the authors should add a reference.
-Since the total content of S. aurea is a quality parameter, why did the author not determine de UFC/ml of milk?
-Line 143: Convert 0.2 μg/L in ng/kg bw so that readers can compare results.
-Line 144: The sentence is confused: “ The OTA levels of 22.5% (27/120) pasteurized milk found in our study are adequate to lead to a higher intake of OTA than the proposed TDI of 5 ng/kg bw in children consuming large amounts of milk.”
- At no point do the authors discuss the co-occurrence of S. aureus and OTA, and they do not explain how they analyzed their co-occurrence. For this reason, I believe that the title and the proposed objectives mislead the reader.
-Conclusion should be rewritten and respond to the objective of the work
Author Response
Manuscript ID: toxins-1916153
The line numbers mentioned below refer to those in the revised manuscript. Point by point responses to editor/reviewers:
Reviewer #4:
We are very grateful to the Reviewer #4 for giving us very professional and specific suggestions to revise our manuscript.
Q1. Rewrite phrase “Therefore, S. aureus containing enterotoxin genes in pasteurized milk could present a food safety risk and needs to be continuously monitored.”-Therefore, pasteurized milk could be a risk due to the occurrence of toxigenic strains of S. aureus
Thanks for the suggestion. We have made changes in the manuscript based on your suggestion. (At line 11-12 & line 286-267). Thanks again for your suggestion.
Q2. I believe the MS should be proofread.
Thanks for the suggestion. I have made additions to the conditions of the mass spectra based on the references. These include:Entrance Potential (EP)、Collision Energy (CE)、precursor ions, product ions, ion ratio, collision energy, De-clustering Potential (DP), Exit Potential (CXP) and so on.(At line 380-383).[41].
Q3. Line 46: Correct 100 ºC… 121 ºC
Thanks for the suggestion. After checking the references, it was confirmed that the values of 100℃ and 121℃ were fine. (At line 81-82)
Q4. Line66: I think authors could explain better the necessity of monitoring OTA.
Thanks for the suggestion. Several necessities of monitoring OTA have been added in the manuscript. Firstly, the toxicity of the OTA itself and its difficult to remove; Second, few researches have been done on OTA detection in milk, however, some studies have shown OTA contamination in milk at relatively high level. Lastly, the fact that the detection of OTA in milk has not been taken seriously. Therefore, monitoring for OTA in milk is necessary. (At line 105-109)
Q5. The authors could restructure the introduction section, adding paragraphs.
Thanks for the suggestion. We have restructured the introduction section and added paragraphs, and the logic has been enhanced. The idea of the preamble is as follows: paragraph 1 talks about the nutritional of milk, its high sales volume, and the risk of biohazards in milk such as S. aureus and OTA. Paragraph 2 talks that milk is one of the important food sources of staphylococcal food poisoning. Paragraph 3 talks about the significance of enterotoxin genetic testing and the hazards of enterotoxin. Paragraph 4 talks about he toxicity of OTA and the difficulty of removing it. OTA can enter the milk through the cow's metabolism and endanger human health. The above is the structure of the whole introduction.
Q6. Avoid references that are 10 years old.
Thanks for the suggestion. The references from the last decade has been replaced in the manuscript with references from the last 5 years whenever possible. There are also some classical references that have not been replaced and are more important sources of information, so they have been retained
Q7. Line 93 how the authors determine LOQ?
Thanks for the suggestion. We determined LOD and LOQ by using Agilent 1260 Infinity Quaternary LC system and a triple-quadrupole mass spectrometer (Agilent Technologies 6460, CA, U.S.A.). The signal-to-noise (S/N) approach was used to estimate the LOD and the LOQ. The gradient OTA standard solution was injected on the liquid chromatograph. The LOD and LOQ were defined based on signal (S)-to-noise (N) ratios of S/N > 3 and S/N > 10, respectively. And the methods for the determination of LOD and LOQ as well as the results have been added to the manuscript. (At line 384-387)
Q8. Table 2: Include information about contamination of each 31 sample.
Thanks for the suggestion. Table 2: has included information about contamination of each 31 sample.It is mainly designed to present the distribution of the 31 contaminated samples in several different OTA contamination level intervals to better highlight the number of samples with higher contamination levels.(At line 155)
Q9. Figure 3: the quality of the diagram is questionable. Perhaps a table could be more explanatory.
Thanks for the suggestion. We previously considered using a table format, but after comparing it with a graph, we came up with a more intuitive and easy to understand graph. Table 3 presents the distribution of the 31 contaminated samples among the different months of the year and the specific OTA detections. The graphical format provides a better visualization of the months with the highest contamination levels and the differences between the months.(At line 160)
Q10. Since the data is from December 2014 to September 2015, I think the authors should explain why they believe the results are still relevant to the present time. Do they think the situation has changed after 7 years? Include this information in the discussion section.
Thanks for the suggestion. First, according to the available references, OTA is known to be present in milk and OTA contamination was detected in liquid bulk milk in Sudan in 2009 at a level of 2.73 µg/ml. But few studies detected OTA in milk. Considering the toxicity of OTA itself and the difficulty of its removal, the objective of this manuscript was to detect OTA in pasteurized milk in Beijing. The results of this manuscript are mainly intended to serve as a warning and to call attention to the detection of OTA in milk. (At line 266-267)
Several studies have examined the prevalence of OTA in different kinds of milk – raw, pasteurised and UHT milk((FloresFlores and González-Peñas 2018; Kamal et al. 2019; Younis et al. 2016; Tale Hel Abad et al. 2016; Turkoglu and Keyvan 2019; Singh and Mehta 2020) (The index of references is listed below) In this references, entitled “Determination of Aflatoxin M1 and Ochratoxin A in Milk and Dairy Products in Supermarkets Located in Mansoura City, Egypt”, the range of OTA levels detected in the four milk samples was 0.34-13 µg/L. The detection rate was 80% (32/40). The detection level of OTA in this references was similar to our results. But the detection rate was higher than ours. And several other papers have smaller values of OTA detection. So after 7 years, OTA in milk have not received more attention relative to the previous situation and there are still milk samples with high levels of OTA. Therefore, after a 7-year interval, the study in this manuscript is still of cautionary significance. (At line 242-250)
The index of the newly added references is as follows:
- Flores-Flores ME, González-Peñas E. 2018. Analysis of mycotoxins in peruvian evaporated cow milk. Beverages. 4:34. doi:10.3390/beverages4020034.
- Kamal R, Mansour M, Elalfy M, Galala W. 2019. Quantitative detection of aflatoxin M1, ochratoxin and zearalenone in fresh raw milk of cow, buffalo, sheep and goat by UPLC XEVO–TQ in Dakahli Governorate, Egypt. J Vet Med Health. 3(1):1–5. doi:10.23880/oajvsr-16000181.
- Younis G, Ibrahim D, Awad A, El-Bardisy MM. 2016. Determination of aflatoxin M1 and ochratoxinA in milk and dairy products in supermarkets located in Mansoura City, Egypt. J Anim Vet Ad. 4:114–121. doi:10.14737/journal.aavs/2016/4.2.114.121.
- Tale Hel Abad S, Joshaghani HR, Rahimzadeh H, Niknejad F. 2016. Ochratoxin A in cow’s milk collected from cattle farms of Golestan province. Med Lab J. 10(1):13–16. doi:10.18869/acadpub.mlj.10.1.13.
- Turkoglu C, Keyvan E. 2019. Determination of aflatoxin M1 and ochratoxin A in raw, pasteurized and UHT milk in Turkey. Acta Sci Veter. 47:1626. doi:10.22456/1679-9216.89667.
- Singh J, Mehta A. 2020. Rapid and sensitive detection of mycotoxins by advanced and emerging analytical methods: a review. Food Sci Nutr. 8(5):2183–2204. doi:10.1002/fsn3.1474.
Q11. line 102: It is a logical sentence, but the authors should add a reference.
Thanks for the suggestion. Detailed data were placed in the response script as an attachment. (At line 221-225)
Q12. Since the total content of S. aurea is a quality parameter, why did the author not determine de UFC/ml of milk?
Thanks for the suggestion. We tested S. aureus in pasteurized milk samples and the detection rate was 99.17% (119/120). The colony count of S. aureus in the sample ranged from 0-100 CFU/mL. Detailed data were placed in an supplementary materials. It is not mentioned in the manuscript because the focus was put on the detection of S. aureus enterotoxin gene.
Q13. Line 143: Convert 0.2 μg/L in ng/kg bw so that readers can compare results.
Thanks for the suggestion. But,this part of the content has been deleted.
Q14. Line 144: The sentence is confused: “ The OTA levels of 22.5% (27/120) pasteurized milk found in our study are adequate to lead to a higher intake of OTA than the proposed TDI of 5 ng/kg bw in children consuming large amounts of milk.”
Thanks for the suggestion. This part of the content has been deleted.
Q15. At no point do the authors discuss the co-occurrence of S. aureus and OTA, and they do not explain how they analyzed their co-occurrence. For this reason, I believe that the title and the proposed objectives mislead the reader.
Thanks for the suggestion. We can understand what you mean by co-occurrence. This manuscript also reflects the co-occurrence of S. aureus and OTA. Co-occurrence means that S. aureus and OTA can be detected in the same milk.The results showed that 29 strains of S. aureus were isolated from 210 samples of pasteurized milk, with a maximum of 100 CFU/ml of total S. aureus , and 17.24% had the sea gene encoding enterotoxin A. The detection rate of OTA in 120 samples was 25.83%, with a maximum detection rate of 18.8 μg/mL Therefore, it is appropriate to use the expression of co-occurrence. Relatively speaking, some of the manuscripts only detected S. aureus or OTA, but we detect both biohazards at the same time .
Q16. Conclusion should be rewritten and respond to the objective of the work
Thanks for the suggestion. It has been rewritten in the manuscript. (At line 285-290). Thank you again !

Round 2
Reviewer 4 Report
The authors have corrected the MS and I now believe it is suitable for publication after minor changes.
Minnor comments:
Correct the order of figures and tables.
Author Response
Thanks for the suggestion. The order of figures and tables(Including supplementary materials) has been corrected according to the content and logic of the article.